# Sustainable Development Economic Strategy Model for Reducing Carbon Emission by Using Real Options Approach

**Chuan-Chuan Ko [1],\* , Chien-Yu Liu [2],\*, Zan-Yu Chen [1] and Jing Zhou [1]**

[1] Department of Business School, City College of Dongguan University of Technology,
Dongguan 523419, China; czy@ccdgut.edu.cn (Z.-Y.C.); zhoujing@ccdgut.edu.cn (J.Z.)

[2] Department of Food and Beverage Management, Jinwen University of Science and Technology,
New Taipei City 23154, Taiwan

\* Correspondence: kejj@ccdgut.edu.cn (C.-C.K.); liuvincent@just.edu.tw (C.-Y.L.)

**Abstract:** This paper is aimed at the call of the United Nations Intergovernmental Panel on Climate Change (IPCC) for the need to maintain global warming within a controllable range. The goal is to target carbon emissions to achieve "net-zero" emissions, along with constructing a green energy investment strategy model for firms in response to government's environmental protection policies. The paper uses the real options approach of dynamic investment decision to construct an investment decision model. Considerations include government taxation of carbon emissions, subsidies to reduce carbon emission policies, and incentives for firms to renew their investments in green energy equipment. Assuming that there is uncertainty in government carbon emission taxes and a reduction of carbon emission subsidies, the changes follow the joint geometric Brownian movement. We used this model to solve the optimum of the threshold for carbon emission taxes and of carbon emission reduction subsidies ratio. If carbon emission taxes and carbon emission reduction subsidies ratio are higher than the threshold, a firm suspends investment in green energy equipment because government subsidies are insufficient. If carbon emission taxes and the carbon emission reduction-subsidy ratio are less than or equal to the threshold, then a firm is qualified for the government's subsidies for reducing carbon emissions, and the firm invests in green energy equipment. The results of this study can provide reference for firms to invest in green energy equipment, and for government control of carbon emission policies. This policy can effectively reduce carbon emissions and achieve co-construction, co-governance, and the sharing of innovative social governance patterns. Finally, it can create a win–win situation between the government, firms, and society.

**Keywords:** climate change; carbon emission tax; subsidy policy; real options; social welfare

## 1. Introduction

Proposed by the United Nations in 2018, the world is now at the most critical moment of climate change. This could lead to serious disasters to all of humanity and the entirety of the planet's ecosystem. To keep global warming within a manageable range, carbon pricing is an important solution. The Intergovernmental Panel on Climate Change (IPCC) report states that net emissions must reach a "net-zero" status by 2050. The 2018 Nobel Prize in Economic Sciences was awarded to Professor William D. Nordhaus, who also proposed the concept of carbon pricing, and presented that the most effective way to solve the problem of greenhouse-gas emissions is to uniformly levy a carbon tax on all countries. Therefore, this paper studied the issue of carbon emission taxes and subsidies to reduce carbon emissions. When the government levies carbon emission taxes,

the production costs of firms increase. Firms may pass on production costs to consumers and reduce social welfare. Therefore, the government should consider the issue of subsidies from a social-welfare perspective and reward carbon-reducing firms by encouraging firms to cooperate with governmental environmental-protection policies, fulfill social responsibilities, reduce carbon emissions, and invest in technological innovation by upgrading green energy equipment. Assuming that other conditions are unchanged, government carbon emission tax levies and carbon emission reduction subsidies are the main decision variables. We applied the real options approach evaluate the feasibility of technological innovation and upgrading green energy equipment investment projects for firms to save energy and reduce carbon emissions. Our results also provide government policies of optimally levying carbon emission taxes and carbon emission reduction subsidies.

The real options approach is essentially a strategic "investment or no investment". The real options approach includes the options to defer development, abandon, expand or contract, extend or shorten, scope up or down, compound options, and rainbow options. It considers the time risk factor and the uncertainty of the future, and the decision maker has the ability to respond to the choice. Under this advantage, the real options approach evaluation project investment has become an important method for modern governments and firms [1]. Traditional methods used for the feasibility evaluation of investment projects include the profitability index (PI), internal rate of return (IRR), payback period, accounting rate of return (ARR), and net present-value methods (NPV). However, in a complex and uncertain investment environment, managerial investment strategies should adopt a dynamic decision-analysis model. Among them, the real options approach is more responsive to a complex investment environment than a traditional NPV [2,3]. Hazra et al. [4] pointed out that the value of mining is affected by uncertain parameter values. The traditional deterministic NPV method in many cases cannot provide the required solution because NPV does not consider uncertain parameters and dynamic properties. Real option valuation (ROV) is a more practical way to solve this problem. Trigeorgis and Reuer [5] reviewed real options theory (ROT) in strategy. ROT can provide managers with the flexibility to deal with the relationship between competition and cooperation in an uncertain environment, and how ROT provides information for managers on key tensions between commitment and resilience, and between competition and cooperation. This theory proposes how to uniquely solve basic problems in the strategy. Smit and Trigeorgis [6] integrated real options theory and game theory to propose a "strategic net present value" theory. Exploring the new value-assessment method that combines real options theory and game theory in an uncertain environment, such as with learning-experience effects, technical uncertainty, and proprietary-information interaction, the optimal choice is in the elastic strategy of waiting and execution. Providing a strategic NPV increases our understanding of conditions that are more relevant to NPV, real options or strategic thinking. Savolainen et al. [7] used the real options approach to study the impact of financing conditions on management flexibility and project value. Liu and Wang [8] developed a network-equilibrium model using real put-and-call option theory to study competitive supply-chain companies that could strategically invest in new supplier capabilities under uncertain cost and demand. Guj and Chandra [9] pointed that using real options analysis approaches avoids the use of positive biases inherent in volatility estimates, which can aggregate the effects of all sources of uncertainty and, to some extent, avoid modeling complexity. Depending on the probability distribution of individual uncertain variables rather than a summary form of cash-flow fluctuations, a more accurate and often more conservative real option value is generated. Ko et al. [10] used a compound binomial options method with management flexibility, considering that cyclical changes in the overall economy affect consumer purchasing power. Gross domestic product (GDP) represented by future economic growth is uncertain. Thinking about product-cycle characteristics, optimal investment strategy, and decision-making project and option values at each stage need to be evaluated. Dolan et al. [11] recommended that the Brennan–Schwartz method be extended to real option valuation to address the issue of physical climate risks faced by companies for long-term investors, such as sovereign wealth funds. Delaney [12] used the real options approach to derive a dynamic model that provides the best timing strategy for trading

techniques. Huberts et al. [13] adopted real options games, considering optimal timing and investment ability under the demand of random dynamic changes. They found that the incumbent invests earlier than the entrant, and that entry deterrence is achieved through timing rather than through overinvestment. Lawrence et al. [14] studied the challenge of decision makers in coastal areas on how to cope with the effects of sustained and uncertain sea-level rise. Dynamic adaptive pathway planning (DAPP) and real options analysis can support decision makers in addressing irreducible uncertainties in coastal areas.

The real options approach applies to environmental pollution and investment to protect environmental issues. Lin et al. [15] used the real options approach to construct a continuous environmental-pollution policy model to establish storage thresholds for contaminated storage and the best time to implement environmental-pollution policies. Liu et al. [16] applied the real options approach to construct an optimal carbon-trading strategy model by applying clean-energy strategy with an optimal energy ratio under clean-energy policy of whether energy is not clean enough or excessively clean. Torani et al. [17], focusing on the development of clean-energy technologies, used a stochastic dynamic model of the real options approach to explore changes in solar technology. After evaluating energy, electricity prices, innovation subsidies, carbon taxes, the optimal investment threshold, and the timing of firms when electricity prices and solar energy costs are uncertain, they proposed policy development as the main key factor. Moriarty and Palczewski [18] used the real options approach to assess the value of power system reserve capacity for a limited period of time. Ansaripoor and Oliveira [19] used the real options approach of management flexibility to construct models that calculate uncertain prices, fuel prices and consumption, and technological advances, and analyze how uncertain fuel prices and technological advances produce choices. Kim et al. [20] studied the economic impact of climate change in Cambodia and determined the economics and feasibility of adaptation strategies such as irrigation and planting-date adjustments by using an investment model based on a real options framework. Providing an appropriate set of policy recommendations can lead to sustainable agricultural productivity and economic growth.

With regard to research on issues related to carbon emission tax and policy, Barbosa et al. [21] developed an extended real options model that took into account some of the relevant macroeconomic factors that were not present in the relevant literature (namely, different types of taxes, asymmetric investment multipliers, and public inefficiencies). The best incentives for different types of stimuli were derived and discussed. They also showed that subsidy policies were always better than tax reductions. Pereira et al. [22] used the dynamic general-equilibrium model to explore the impact of carbon taxes on Portugal. The results showed that carbon-tax revenues give back to lower tax burdens and improve energy efficiency. Whitford [23] pointed out that a major impact of U.S. energy-price changes is political structural concerns about oil and gas markets and government support for energy efficiency. Voulis et al. [24] explored the potential of energy taxes to provide incentives. Through measuring the difference in financial incentives between two tax designs (per-unit and ad valorem taxes) in a simulation case study of consumers' heat pumps in the Netherlands, their outcomes showed that financial incentives were 3.5 times higher for the ad valorem tax than for the per-unit tax. They also recommended that energy-tax policy should be formulated to provide consumers with adequate financial incentives. Ma et al. [25] pointed out that, with the rapid development of their economy, China's power consumption has dramatically increased. They used structural analysis of input–output subsystem analysis in research to explore the sources of emission increases in China's power industry from 2007 to 2015 and further assess the impact of power structures and carbon-tax collection. The research results showed that consumption is a major growth factor for $CO_2$ emissions. Most $CO_2$ emissions are driven by the continued expansion of large-scale infrastructure. Furthermore, carbon taxes and price policies may be an alternative to reducing $CO_2$ emissions. Zhang and Zhang [26] studied the impact of GDP, trade structure, exchange rate, and FDI inflows on China's carbon emissions from 1982 to 2016. They found that the Environmental Kuznets Curve (EKC) hypothesis is valid for China and, at the same time, affects service trade, that China's carbon emission

exchange rates are negative, and that the impact of FDI inflows is positive. Yun et al. [27] explored effective technology-development strategies for solar companies facing technological turbulence (diversification and collaboration). This study found that the adoption of technology diversification and R and D cooperation strategies have a positive impact on business performance. This paper is mainly aimed at firms to improve their environmental responsibility, cooperate with government policies to strengthen environmental protection, and carry out innovative production technologies and pollution-reduction investment projects. Liu [28] studied that, in low-carbon economic networks, governments and enterprises inevitably encounter some degree of mistrust because of the complexity and uncertainty of policies. Research results indicated that an open policy process, joint work, and information sharing are effective methods that reduce the level of distrust. In addition, specific low-carbon policies and low-carbon product standards need to be specifically formulated to reduce the corporate distrust of low-carbon policies. Bryant et al. [29] pointed out that in the transition to renewable-energy, if there is no viable business model to support it, the best renewable technologies and policies decrease short of the transition necessary to develop a sustainable-energy sector.

Section 2 of this paper constructs a decision-making model for innovative energy-saving, carbon-reduction technologies, and equipment purchases under the assumption that the government levies carbon emission taxes and introduces carbon emission reduction subsidies in accordance with the joint geometric Brownian motion hypothesis. Section 3 applies numerical-example and sensitivity analyses. Finally, Section 4 is our conclusion.

## 2. Model

This paper considers that the government levies carbon emission taxes on firms to control carbon emissions under continuous time application and uncertain carbon emissions in order to prevent firms from passing the cost of taxation to consumers, resulting in a decline in social welfare. At the same time, the government subsidizes firms that reduce carbon emissions, and encourages firms to invest in technological innovation and upgrading green energy equipment that reduces carbon emissions. According to the social-welfare point of view, the government must levy firms' carbon emission taxes to reduce carbon emissions, and provide a subsidy policy for firms that effectively reduce carbon emissions. This study applies the real options approach of dynamic investment strategy to construct an optimal decision-making model for firms to respond to government carbon emission taxes and carbon emission reduction subsidy policies.

### 2.1. Assumptions

This paper assumes that carbon emissions generated by firms during the production process are levied on carbon emission taxes by the government. On the other hand, the government implements a subsidy policy to reward firms for investing in technological innovation and upgrading green energy equipment to reduce carbon emissions. Under the risk-neutral assumption, firms invest in innovative technologies and upgrade green energy equipment. The profit function is outlined as Equation (1):

$$f(P_e(t), P_c(t)) = R - Q_{2e}P_e(t) + (Q_{1e} - Q_{2e})P_c(t),　　　　　(1)$$

where $R$, is revenue minus variable costs when firms do not consider costs associated with carbon emissions; $Q_{1e}$, carbon emissions generated before the production equipment has been upgraded; $Q_{2e}$, amount of carbon emissions after upgrading production equipment.; $(Q_{1e} - Q_{2e})$, carbon emissions reduced after technological innovation and upgrading green energy equipment; $P_e(t)$, government-to-firms levy per unit of carbon emission tax; and $P_c(t)$, subsidy for each unit of reduction in carbon emissions, which is the government's subsidy for firms to reduce the effectiveness of carbon emissions. If firms invest in technological innovation and upgrading green energy equipment, input fixed cost is $I_g$. To maintain a controllable range for global warming and control carbon emissions quickly and effectively, assuming the government is targeting uncertain global warming, the firms'

carbon emission tax per unit $P_e(t)$ and the subsidy for reducing carbon emissions per unit $P_c(t)$ are uncertain variables. $P_e(t)$ and $P_c(t)$ changes follow the joint geometric Brownian motion:

$$dP_e(t) = \alpha_e P_e(t)dt + \sigma_e P_e(t)dZ_e(t) \tag{2}$$

$$dP_c(t) = \alpha_c P_c(t)dt + \sigma_c P_c(t)dZ_c(t), \tag{3}$$

where $\alpha_e$, expected growth rate of $P_e(t)$; $\alpha_c$, expected growth rate of $P_c(t)$; $\sigma_e$ and $\sigma_c$, standard deviation of $P_e(t)$ and $P_c(t)$; and $dZ_e(t)$ and $dZ_c(t)$, increments of the standard Wiener process. Here, $\mathrm{E}[dZ_e(t)] = 0$, $\mathrm{E}[dZ_e(t)]^2 = dt$, $\mathrm{E}[dZ_c(t)] = 0$, $\mathrm{E}[dZ_c(t)]^2 = dt$, and $\mathrm{E}[dZ_e(t) \times dZ_c(t)]^2 = \gamma_{e,c}\sigma_e\sigma_c dt$. $\gamma_{e,c}$ are related factors $-1 \le \gamma_{e,c} \le 1$. The expected income of the firm's production operations considers the cost of carbon emissions and fixed costs $I_g$ of inputs such as Equation (4):

$$\mathrm{E}\left[\int_0^\infty R \times e^{-rt}dt - Q_{2e} \times P_e(t) \times e^{-rt}dt + (Q_{1e} - Q_{2e}) \times P_c(t) \times e^{-rt}dt\right] - I_g \tag{4}$$

Factor $r$ is the discount rate. The first item is the expected future cash inflow of firms without considering the cost of carbon emissions. The second item is carbon emissions from the production process of the firms according to the expected future cost of the government levying carbon emission taxes on the firms. The third item is the effectiveness of firms in reducing carbon emissions, whereby government subsidies for carbon emissions present value. The fourth item is the fixed cost of a firm's investment in technological innovation and upgrading green energy equipment. From Equation (4), the net present value of the expected income of firms is as shown in Equation (5):

$$\pi(P_e(t), P_c(t)) = \frac{R}{r} - I_g - \left(\frac{Q_{2e} \times P_e(t)}{r - \alpha_e} - \frac{(Q_{1e} - Q_{2e}) \times P_c(t)}{r - \alpha_c}\right) \tag{5}$$

In Equation (5), the net present value of the expected income of firms is decomposed into the present value of the profit without reducing the cost of carbon emissions, minus the fixed cost, and minus the government levies of carbon emission taxes and government subsidies.

The present value of government levies on carbon emission taxes and the subsidies to reduce carbon emissions of the expected tax is $T(P_e(t), P_c(t))$, such as in Equation (6):

$$T(P_e(t), P_c(t)) = \frac{Q_{2e} \times P_e(t)}{r - \alpha_e} - \frac{(Q_{1e} - Q_{2e}) \times P_c(t)}{r - \alpha_c} \tag{6}$$

From Equation (6), the first item is the present value of government levies on firms' carbon emission taxes. The second item is the present value of government subsidies for firms to reduce carbon emissions. $T(P_e(t), P_c(t))$ is also the present value of the net cost of carbon emissions from firms during the production process.

## 2.2. Decision Model

From the perspective of overall social welfare, the government levies a per-unit carbon emission tax $P_e(t)$ to reduce carbon emissions. In order to prevent firms from transferring increased costs to consumers, the government enacts standards to subsidize per unit $P_c(t)$ of carbon emissions for firms that effectively reduce carbon emissions. Then, $P_e(t)$ and $P_c(t)$ are related. $V(P_e(t), P_c(t))$ is the impact value of the firm's renewal of technological innovation and upgrading green energy equipment, under the government's levy and subsidy of carbon emission pollution-policy strategy. Using Itô's Lemma theorem [30], its government-policy value of management flexibility is as follows Equation (7):

$$\begin{aligned}dV(P_e(t), P_c(t)) = \ &V_{P_e}(P_e(t), P_c(t))dP_e + V_{P_c}(P_e(t), P_c(t))dP_c \\ &+ \tfrac{1}{2}\left[V_{P_eP_e}(P_e(t), P_c(t))(dP_e)^2 + 2V_{P_eP_c}(P_e(t), P_c(t))dP_eP_c + V_{P_cP_c}(P_e(t), P_c(t))(dP_c)^2\right]\end{aligned} \tag{7}$$

Whereby $V_{P_e}(P_e(t), P_c(t))$, $V_{P_c}(P_e(t), P_c(t))$ and $V_{P_e P_e}(P_e(t), P_c(t))$, $V_{P_c P_c}(P_e(t), P_c(t))$ are the first- and the second-order differential equations derived from $V(P_e(t), P_c(t))$ for $P_e(t)$, $P_c(t)$. The expected return over an interval time $dt$, $rV(P_e(t), P_c(t))dt$, is equal to its expected potential value based on the conditions of risk discount rate $r$. The Bellman Equation [3] is as Equation (8):

$$rV(P_e(t), P_c(t))dt \;=\; E[dV(P_e(t), P_c(t))] \tag{8}$$

Replacing Equations (2) and (3) with Equations (7) and (8), the value of the management flexibility of government-policy strategy is as Equation (9):

$$\frac{1}{2}\sigma_e^2 P_e^2 V_{P_e P_e}(.) + \gamma_{e,c}\sigma_e\sigma_c V_{P_e P_c}(.) + \frac{1}{2}\sigma_c^2 P_c^2 V_{P_c P_c}(.) + \alpha_e P_e V_{P_e}(.) + \alpha_c P_c V_{P_c}(.) - rV(.) \;=\; 0 \tag{9}$$

The general solution of Equation (9) is $V(P_e(t), P_c(t)) \;=\; A P_e(t)^{\beta} P_c(t)^{1-\beta}$ [31,32]. The management-flexibility value of government-policy strategy seen by substitution, $\beta$ as a root of the quadratic equation satisfies the equation provided:

$$\frac{1}{2}\sigma^2 \beta(\beta - 1) + \beta(\alpha_c - \alpha_e) - (r - \alpha_e) \;=\; 0 \tag{10}$$

By ordering $\sigma^2 \;=\; \sigma_e^2 - 2\gamma_{e,c}\sigma_e\sigma_c + \sigma_c^2$, the two roots are:

$$\beta_1 \;=\; \frac{\left(\frac{1}{2}\sigma^2 + \alpha_e - \alpha_c\right) + \sqrt{\left(\frac{1}{2}\sigma^2 + \alpha_e - \alpha_c\right)^2 + 2\sigma^2(r - \alpha_e)}}{\sigma^2} \;>\; 1 \tag{11}$$

$$\beta_2 \;=\; \frac{\left(\frac{1}{2}\sigma^2 + \alpha_e - \alpha_c\right) + \sqrt{\left(\frac{1}{2}\sigma^2 + \alpha_e - \alpha_c\right)^2 + 2\sigma^2(r - \alpha_e)}}{\sigma^2} \;<\; 0 \tag{12}$$

The general solution can be written as $V(P_e(t), P_c(t)) \;=\; A_1 P_e(t)^{\beta_1} P_c(t)^{1-\beta_1} + A_2 P_e(t)^{\beta_2} P_c(t)^{1-\beta_2}$, where $A_1$ and $A_2$ are constants to be determined. $V(P_e(t), P_c(t))$ must satisfy the following boundary conditions:

$$\lim_{P_e(t) \to 0} V(P_e(t), P_c(t)) \;=\; 0 \tag{13}$$

Equation (13), when boundary condition $P_e(t)$ goes to zero, stays at $V(P_e(t), P_c(t)) \;=\; 0$. The solutions must take the following form:

$$V(P_e(t), P_c(t)) \;=\; A_1 P_e(t)^{\beta} P_c(t)^{1-\beta} \tag{14}$$

To solve optimal carbon emission tax per unit $P_e(t)$ and reduce carbon emission subsidy emission per unit $P_c(t)$ based on the value-matching and smoothing conditions on threshold $P_e(t)^*$, $P_c(t)^*$, the value of management-elasticity strategy $A_1 P_e(t)^{\beta} P_c(t)^{1-\beta}$ is equal to the government-levy carbon emission tax and subsidies to reduce carbon emissions of the expected tax net present value $T(P_e(t), P_c(t))$. This value-matching condition satisfies the value-uniqueness condition, such as in Equation (15). Then, the marginal value is equal in the first-order derivative function, that is, it satisfies the conditions of equal marginal value. This is a smooth-pasting condition, such as in Equations (16) and (17) [3]:

$$V(P_e(t)^*, P_c(t)^*) \;=\; T(P_e(t)^*, P_c(t)^*) \tag{15}$$

$$\frac{\partial V(P_e(t)^*, P_c(t)^*)}{\partial P_c(t)^*} \;=\; \frac{\partial T(P_e(t)^*, P_c(t)^*)}{\partial P_c(t)^*} \tag{16}$$

$$\frac{\partial V(P_e(t)^*, P_c(t)^*)}{\partial P_e(t)^*} \;=\; \frac{\partial T(P_e(t)^*, P_c(t)^*)}{\partial P_e(t)^*}, \tag{17}$$

where $P_e(t)$, $P_c(t)$ are satisfied by value function $V(P_e(t), P_c(t))$ that is linearly homogeneous. Then, $v(H) = \frac{V(P_e(t), P_c(t))}{P_c(t)}$, $w(H) = \frac{T(P_e(t), P_c(t))}{P_c(t)}$, order $H = \frac{P_e(t)}{P_c(t)}$ [32,33], $H$ indicates the actual ratio of government levies on firms' carbon emission taxes divided by subsidies for reduced carbon emissions. Therefore, Equations (15–17) can be expressed as:

$$\lim_{H \to 0} v(H) = 0 \tag{18}$$

$$v(H^*) = w(H^*) \tag{19}$$

$$\frac{\partial v(H^*)}{\partial H} = \frac{\partial w(H^*)}{\partial H} \tag{20}$$

$H^* = \frac{P_e(t)^*}{P_c(t)^*}$ is the optimal threshold when firms invest in technological innovation and upgrading green energy equipment indicated by the ratio of government levies to firms' carbon emission tax per unit $P_e(t)$ divided by subsidies for firms' carbon emissions reduced per unit $P_c(t)$. The value function of firms' upgraded production equipment to social welfare is as Equation (21):

$$F(H) = \begin{cases} -A_1 H^{\beta_1}, H > H^* \\ \frac{R}{r} - I_g - \left[ \frac{Q_{2e} \times H}{r - \alpha_e} - \frac{(Q_{1e} - Q_{2e})}{r - \alpha_c} \right], H \leq H^* \end{cases} \tag{21}$$

Equation (21), the social-welfare value function includes the management-flexibility value of government-levy and -subsidy carbon emission tax $A_1 H^{\beta_1}$ for firms. Because the government levies and subsidizes the carbon emission tax for firms to suppress carbon emissions, it is an additional cost for firms to invest in green energy equipment, so it is $-A_1 H^{\beta_1}$ at Equation (21). Revenue minus variable costs present value, not considering the costs associated with carbon emission $R$, minus fixed cost $I_g$, minus government levies on carbon emission tax and subsidies to reduce carbon emissions of expected tax net present value $\frac{Q_{2e} \times H}{r - \alpha_e} - \frac{(Q_{1e} - Q_{2e})}{r - \alpha_c}$. A firm's investment in technological innovation and upgrading green energy equipment is determined by the government's policy of controlling carbon emissions. When $H > H^*$, it indicates that the government has set a carbon emission tax per unit $P_e(t)$ that is too high, and that the reduction of carbon emission subsidies per unit $P_c(t)$ is too small. Firms adopt a conservative strategy and wait for a better time to invest in technological innovation and upgrading green energy equipment. When $H \leq H^*$, the firms should choose to update investment decision to technological innovation and upgrading green energy equipment, and the government's net tax is $\frac{Q_{2e} \times H}{r - \alpha_e} - \frac{(Q_{1e} - Q_{2e})}{r - \alpha_c}$. Solving $H^*$ and $A_1$ by Equation (21) by using value-matching condition and smooth-pasting conditions is shown in Equation (22):

$$\begin{cases} -A_1 H^{*\beta_1} = \frac{R}{r} - I_g - \left[ \frac{Q_{2e} \times H^*}{r - \alpha_e} - \frac{(Q_{1e} - Q_{2e})}{r - \alpha_c} \right] \\ A_1 \beta_1 H^{*\beta_1 - 1} = \frac{Q_{2e}}{r - \alpha_e} \end{cases} \tag{22}$$

Using Equation (22), this yields $H^*$ and $A_1$ as:

$$H^* = \frac{\beta_1}{(\beta_1 - 1)} \times \frac{(r - \alpha_e) \times \left[ R \times (r - \alpha_c) + (Q_{1e} - Q_{2e}) \times r - I_g \times r \times (r - \alpha_c) \right]}{Q_{2e} \times r \times (r - \alpha_c)} \tag{23}$$

$$A_1 = \frac{Q_{2e} \times H^{*1 - \beta_1}}{\beta_1 (r - \alpha_e)}, \tag{24}$$

where $H^* = \frac{P_e(t)^*}{P_c(t)^*}$ is the optimal threshold when firms invest in technological innovation and upgrading green energy equipment indicated by the ratio of the government's levies to firms' carbon emission tax

per unit $P_e(t)^*$ divided by subsidies for firms' carbon emissions reduced per unit $P_c(t)^*$. If governments levy carbon emission tax per unit $P_e(t)^*$ and reduce the carbon emission subsidy per unit $P_c(t)^*$ ratio, $H$ is higher than $H^*$. It indicates that the government has levied too high a carbon emission tax on firms, and the government has insufficient subsidies for firm investment in reducing carbon emissions. Therefore, firms choose to suspend investment in green energy equipment and wait for better investment opportunities. If $H$ is less than or equal to $H^*$, it indicates that the government has levied an emission tax on firms and, at the same time, sufficient subsidies for firms to reduce emissions. Firms choose to invest in technological innovation and upgrading green energy equipment.

This paper constructed an investment-decision model to solve optimal threshold $H^*$. The threshold is the government's goal of protecting the environment and reducing carbon emissions. When the government levies a carbon tax on firms they need to ensure two things: avoid the transfer of carbon emission costs to consumers, and award carbon emission subsidy to firms that have reduced carbon emissions. This paper constructed a model focusing on how to set taxes and subsidies that can encourage firms to choose technological innovation and upgrading green energy equipment investment, and effectively control carbon emissions. Optimal threshold $H^*$ is the reference for the government to formulate the carbon emission tax for firms and the policy of reducing carbon emission subsidies for firms, and the timing for firms to select technological innovation and upgrading green energy equipment.

## 3. Numerical-Example and Sensitivity Analysis

This section conducts numerical-example and sensitivity analysis for the decision model constructed in Section 2 by exploring how the government can effectively control carbon emissions under the pursuit of economic growth and environmental-protection issues.

### 3.1. Numerical Example

Zhou et al. [34], based on China's 1995–2014 statistics, found that carbon emissions have a significant effect on energy consumption and economic growth. Results showed that carbon emissions contribute more to fluctuations of economic growth and energy consumption, which indicated that China's unit GDP produced higher carbon emissions. So, according to historical data, fluctuations in annual GDP growth rate in China between 2008 and 2017 would have affected the change in carbon emissions that, in turn, would cause the government to formulate a firm carbon emission tax strategy. This paper used GDP data to estimate the expected growth rate of the carbon emission tax. Then, the discount rate was mainly the expected return rate of the firms. The paper estimated the rate of return on common stockholder equity of listed Chinese companies from 2000 to 2016 by about 15% [35]. We suggest that governments should not only levy carbon emission taxes, but also adopt subsidies on firms that effectively reduce carbon emissions, assuming that the expected growth rate of subsidies is less than the expected growth rate of carbon emission tax. This section uses numerical examples for analysis. According to the decision model of this paper, we solved optimal threshold $H^*$ for a government's carbon emission tax levy per unit $P_e(t)$ and carbon emission reduction subsidies per unit $P_c(t)$ ratio. $H^*$ is the optimal threshold for the government to formulate policies and firms to invest in green energy production equipment. Their associated exogenous variables were assumed as shown in Table 1.

Numerical-analysis results derived from exogenous external variables and numerical data in Table 1 showed the optimal threshold for government carbon emission tax levy per unit $P_e(t)$ and carbon emission reduction subsidy per unit $P_c(t)$ ratio $H^* = 0.91$; parameter $A_1 = 77,958.99$. Optimal threshold $H^* = \frac{P_e^*(t)}{P_c^*(t)} = 0.91$ is the reference value for the optimal investment timing of the production equipment for firms. That is, when the threshold for the government carbon emission tax levy per unit $P_e(t)$ and carbon emission reduction subsidy per unit $P_c(t)$ ratio $H \leq H^* = 0.91$, firms should adopt an investment strategy to invest in technological innovation and upgrading green energy equipment. However, even when governments' carbon emission tax levy per unit $P_e(t)$ is less

than carbon emission reduction subsidy per unit $P_c(t)$ under other unchanged conditions, in order to motivate firms to actively cooperate with the government's environmental protection policy before the goal of "net zero" of carbon emissions is achieved, the government should adopt a carbon emission tax per unit less than the carbon emission reduction subsidy per unit policy. Optimal threshold $H^*$ provides reference for a government to formulate policies, and for firms to invest in technological innovation and upgrading green energy equipment.

**Table 1.** Exogenous variables.

| Exogenous Variables | Significance | Value |
|:---:|:---:|:---:|
| $\alpha_e$ | Expected growth rate of carbon emission tax per unit $P_e(t)$. | 0.08 |
| $\alpha_c$ | Expected growth rate for carbon emission reduction subsidies per unit $P_c(t)$. | 0.05 |
| $\sigma_e$ | Standard deviation of carbon emission tax per unit $P_e(t)$. | 0.30 |
| $\sigma_c$ | Standard deviation of carbon emission reduction subsidies per unit $P_c(t)$. | 0.20 |
| $r$ | Risk discount rate | 0.15 |
| $\gamma_{e,c}$ | Correlation coefficient between $P_e(t)$ and $P_c(t)$. | 1.00 |
| $Q_{1e}$ | Carbon emissions before upgrading green energy production equipment (unit: million tons). | 2.00 |
| $Q_{2e}$ | Carbon emissions after upgrading green energy production equipment (unit: million tons) | 1.60 |
| $I_g$ | Upgrading equipment to invest in fixed costs (unit: million dollars). | 6000.00 |
| $R$ | Firm production income not considering carbon emission costs (unit: million dollars). | 2600.00 |

### 3.2. Sensitivity Analysis

Sensitivity analysis was applied to the effects of exogenous variables on optimal threshold of governments carbon emission tax levy per unit $P_e(t)$ and carbon emission reduction subsidy per unit $P_c(t)$ ratio $H^*$. First, with the assumption that the other conditions are unchanged, this paper analyzed changes in risk discount rate $r$ that affected optimal threshold $H^*$. These changes are shown in Table 2:

**Table 2.** Influence of risk discount rate $r$ on $H^*$.

| $r$ (Risk Discounted Rate) | $H^*$ (Optimal Threshold $P_e(t)$ Divided by $P_c(t)$) |
|:---:|:---:|
| 0.14 | 0.87 |
| 0.15 | 0.91 |
| 0.16 | 0.94 |
| 0.17 | 0.96 |
| 0.18 | 0.98 |

.

As shown in Table 2, when risk discount rate $r$ rises, optimal threshold $H^*$ also rises, which means that, if firms intend to increase the rate of return on investment, the government can increase carbon emission tax per unit $P_e(t)$ or decrease carbon emission reduction subsidy per unit $P_c(t)$ when formulating policies. Firms may still have the willingness to cooperate with government environmental-protection policies to invest in technological innovation and upgrading green energy equipment that reduces carbon emissions.

Second, with the assumption that other conditions are unchanged, this paper analyzed the equal change of expected growth rate $\alpha_e$ of carbon emission tax $P_e(t)$ and expected growth rate $\alpha_c$ of carbon emission reduction subsidies $P_c(t)$ that affect optimal threshold $H^*$. Changes are shown in Table 3:

**Table 3.** Influence of expected growth rate $\alpha_e$ and $\alpha_c$ on $H^*$.

| $\alpha_e$ (Expected Growth Rate $\alpha_e$ of Carbon Emission Tax) | $\alpha_c$ (Expected Growth Rate $\alpha_c$ of Carbon Emission Reduction Subsidies) | $H^*$ (Optimal Threshold $P_e(t)$ divided by $P_c(t)$) |
|---|---|---|
| 0.08 | 0.05 | 0.91 |
| 0.09 | 0.06 | 0.81 |
| 0.10 | 0.07 | 0.70 |
| 0.11 | 0.08 | 0.58 |
| 0.12 | 0.09 | 0.46 |

As shown in Table 3, when expected growth rate $\alpha_e$ of carbon emission tax per unit $P_e(t)$ and expected growth rate $\alpha_c$ of carbon emission reduction subsidy per unit $P_c(t)$ have the same increase, optimal threshold $H^*$ declined. This result indicated that governments try to effectively suppress a firm's carbon emissions and hope that firms invest in technological innovation and upgrading green energy equipment that reduces carbon emissions. When governments levy a higher carbon emission tax on firms, they need to provide more subsidies, and when expected growth rate $\alpha_e$ of carbon emission tax per unit $P_e(t)$ reaches $\alpha_e = 12\%$, subsidy per unit $P_c(t)$ of carbon for reducing carbon emissions is more than twice the amount of tax per unit $P_c(t)$.

Third, with the assumption that other conditions are unchanged, this paper analyzed how changes in revenue $R$ from firms not considering carbon emission costs affect optimal threshold $H^*$. Changes are shown in Table 4:

**Table 4.** Influence of revenue $R$ on $H^*$.

| $R$ (Revenue Not Considering Carbon Emission Costs (Unit: million dollars)) | $H^*$ (Optimal Threshold $P_e(t)$ Divided by $P_c(t)$) |
|---|---|
| 25,000.00 | 0.87 |
| 26,000.00 | 0.91 |
| 27,000.00 | 0.93 |
| 28,000.00 | 0.99 |
| 29,000.00 | 1.03 |

As shown in Table 4, when revenue $R$ rises, optimal threshold $H^*$ also rises. Therefore, when the investment revenue of firms is raised, governments should adopt an increase of carbon emission tax per unit $P_e(t)$ when formulating carbon emission tax policies. Meanwhile, when revenue increases to 29,000 million dollars, optimal threshold $H^*$ changes from being smaller than 1 to greater than 1. That is, when revenue reaches more than 29,000 million dollars, governments may formulate carbon emission tax policies by taking a per-unit carbon emission tax greater than per-unit carbon emission subsidies. Firms still have the willingness to cooperate with governments' environmental-protection policies to invest in technological innovation and upgrading green energy equipment that reduces carbon emissions.

Finally, with the assumption that the other conditions are unchanged, this paper analyzed how $Q_{2e}$, representing carbon emissions after firms have invested in renewable green energy production equipment, affects optimal threshold $H^*$. Changes are shown in Table 5:

**Table 5.** Influence of the $Q_{2e}$ on $H^*$.

| $Q_{2e}$ (Carbon Emissions After Firms Upgraded Their Green Energy Production Equipment) | $Q_{1e} - Q_{2e}$ (Reduced Carbon Emissions) | $H^*$ (Optimal Threshold $P_e(t)$ Divided by $P_c(t)$) |
|---|---|---|
| 18,000.00 | 2,000.00 | 0.71 |
| 17,000.00 | 3,000.00 | 0.80 |
| 16,000.00 | 4,000.00 | 0.91 |
| 15,000.00 | 5,000.00 | 1.04 |
| 14,000.00 | 6,000.00 | 1.18 |

As shown in Table 5, when carbon emissions $Q_{2e}$ are decreased, reduction in carbon emissions is more effective, that is, $Q_{1e} - Q_{2e}$ and optimal threshold $H^*$ are greater. From a government's perspective, when formulating policies for firms that are better at reducing carbon emissions, carbon emission tax per unit $P_e(t)$ should be higher than carbon emission subsidy per unit $P_c(t)$, that is, $H^* > 1$. For firms that are not sufficiently reducing carbon emissions, carbon emission tax per unit $P_e(t)$ should be lower than carbon emission subsidy per unit $P_c(t)$, that is, $H^* < 1$. It is helpful to encourage firms toward technological innovation and upgrading green energy equipment as soon as possible. The more the firms that reduce carbon emissions, the higher optimal threshold $H^*$ is. Governments should thus moderately reduce the subsidy per unit of carbon emissions to reduce fiscal expenditure.

## 4. Conclusions

The purpose of the paper was to promote a win–win strategy for governments, firms, and the natural environment under sustainable economic development. The paper was based on reviews of flexible policy management concerning how the government could effectively control environmental-protection issues of carbon emissions. In order to effectively control carbon emissions in the production process, governments levy a carbon tax on firms in order to prevent firms from passing on increased tax cost to consumers. Therefore, incentive firms actively invest in technological innovation and upgrading green energy equipment. When a government levies a carbon emission tax on firms, it also adopts subsidies for firms to invest in technological innovation and upgrading green energy equipment to reward firms that reduce carbon emissions. Therefore, the paper facilitates governments' decisions to levy carbon emission taxes and subsidize the reduction of carbon emissions by creating an optimal decision-making model for firms to invest in technological innovation and upgrading green energy equipment.

The model mainly determines the best timing of firms' investment in technological innovation and upgrading green energy equipment. In response to the government, $CO_2$ net emissions must reach a "net-zero" state. Governments levy a carbon emission tax and subsidize the reduction of carbon emission policies to promote firms' investment in technological innovation and upgrading green energy equipment. This paper assumed that government carbon emission tax levy per unit $P_e(t)$ and carbon emission reduction subsidy per unit $P_c(t)$ are uncertain variables, and their changes follow joint geometric Brownian motion. This paper used the real-option approach to construct a decision-making model for carbon emission policy formulation to derive optimal threshold $H^*$ for carbon emission tax per unit $P_e(t)^*$ and carbon emission reduction subsidy per unit $P_c(t)^*$ ratio. If governments levy carbon emission tax per unit $P_e(t)$ and reduce carbon emission subsidy per unit $P_c(t)$ ratio, $H$ is higher than $H^*$. This indicates that government subsidies are insufficient for firms' investment in reducing carbon emissions. Therefore, firms choose to suspend investment in technological innovation and upgrading green energy equipment, and wait for better investment opportunities. If $H$ is less than or equal to $H^*$, firms choose to invest in technological innovation and upgrading green energy equipment. Based on numerical-example- and sensitivity-analysis results, the theoretical and practical implications of this study are as follows.

Numerical-example analysis showed that the reference value for optimal investment of technological innovation and upgrading green energy equipment is when optimal threshold $H^* = 0.91$. However, when $H$ is less than or equal to 0.91, firms should adopt a strategy of investing in technological innovation and upgrading green energy equipment to reduce carbon emissions. At the same time, it also showed that, when governments start to formulate carbon emission taxes, they should provide higher subsidies on reducing carbon emissions than carbon emission taxes, and further encourage firms to actively invest in technological innovation and upgrading green energy equipment. Sensitivity-analysis results showed that, when risk discount rate $r$ rises, optimal threshold value $H^*$ also rises. However, when firms' investment return rate $r$ is increased, governments can increase carbon emission taxes or decrease subsidies to reduce carbon emissions when formulating policies. Moreover, when carbon emission taxes' expected growth rate $\alpha_e$ and expected growth rate $\alpha_c$ of

carbon emission subsidy increase, optimal threshold $H^*$ decreases. This indicates that governments need to provide more subsidies to reduce carbon emissions when imposing carbon emission taxes on firms. Additionally, when revenue $R$ of firms that have not considered carbon emission costs rises, optimal threshold $H^*$ also rises. Therefore, when the investment revenue of firms is raised, governments should adopt an increase of carbon emission taxes per unit $P_e(t)$. When revenue $R$ increases to a certain extent, optimal threshold $H^*$ is also changed from being less than 1 to greater than 1. That is, governments formulate carbon emission tax policies when per-unit carbon emission taxes are greater than per-unit carbon emission subsidies. As such, optimal threshold $H^*$ increases as carbon emissions $Q_{2e}$ decrease after firms invest in technological innovation and upgrading green energy equipment. From a government's perspective, firms that fail to meet the "net-zero" target should be given higher subsidies in order to promote firms investing in technological innovation and upgrading green energy equipment as soon as possible. The results of this study can provide reference for firms to invest in technological innovation and upgrading green energy equipment, governments to control emission policies, and, in turn, to achieve innovative social-governance patterns of co-construction, co-governance, and sharing, which creates a win–win situation between governments, firms, and society.

This study limits the main research from a financial point of view. This paper assumed that government carbon emission tax levy per unit $P_e(t)$ and carbon emission reduction subsidy per unit $P_c(t)$ are uncertain variables, and their changes follow joint geometric Brownian motion, and only used subsidies to promote firms investing in technological innovation and upgrading green energy equipment. Our recommendations for future research directions are: (1) changes of government carbon emission tax levy per unit $P_e(t)$ and carbon emission reduction subsidy per unit $P_c(t)$ are subject to various random processes according to the actual situation, such as the Poisson process; (2) variables in the model can be expanded to being multivariate; (3) ways to promote investment in technological innovation and upgrading green energy production equipment can be promoted by education, regulation, and other aspects.

**Author Contributions:** conceptualization, C.-C.K., C.-Y.L., Z.-Y.C., and J.Z.; funding acquisition, C.-C.K.; methodology, C.-C.K. and C.-Y.L.; writing—original draft, C.-C.K. and C.-Y.L.; writing—review and editing, C.-C.K., C.-Y.L., Z.-Y.C., and J.Z.

**Funding:** This research was funded by the City College of Dongguan University of Technology (no. 2017YZDYB03R).

**Acknowledgments:** The authors would like to thank the anonymous reviewers for their valuable comments, which helped improve the presentation of this article.

**Conflicts of Interest:** The authors declare no conflict of interest.

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
