# Peer review of "Sustainable Development Economic Strategy Model for Reducing Carbon Emission by Using Real Options Approach"

_sustainability, doi:10.3390/su11195498_

Round 1

Reviewer 1 Report

Dear authors,

It was a pleasure to read your research as it debates and analyses current and critical issues and solutions regarding the tackling of greenhouse gas emissions.

In terms of improving the paper, in my opinion, you should indicate the data sources for all the information required fo the analyses and simulations that you have conducted in this study. Also, if it is the case, you should present the software used for conducting this research, as maybe the decision makers and other stakeholders would like to replicate it. 

In addition, in Conclusions section the phrase "The 2018 Nobel Prize in Economic Sciences was awarded to professor William D. Nordhaus who proposed that the most effective way to solve the problem of greenhouse gas emissions is to uniformly levy carbon emission taxes on all countries."(lines 359-361) could be eliminated as it is the same from the Introduction section and, in my opinion, it is redundant. 

Finally, in my opinion, you should also indicate possible limitation of your study and ways of improving it in the future.

Best wishes,

Alina Zaharia

Author Response

Dear Reviewer,

We are grateful to have the opportunity to submit our revised manuscript to the Sustainability. Please find herewith enclosed our revised manuscript entitled “Sustainable Development Economic Strategy Model for Reducing Carbon Emissions by Using Real Options Approach” and reviewer reports (point by point), highlights version and cleaned version with coauthored Chien-Yu Liu, Zan-Yu Chen and Jing Zhou. We greatly appreciate if you could arrange to review our manuscript.

Hope you would very kindly acknowledge the receipt of the revised manuscript.

With our best wished and regards.

Sincerely yours,

The authors

Chuan-Chuan Ko, Chien-Yu Liu, Zan-Yu Chen and Jing Zhou

Reviewer 2 Report

The subject under study is very current and relevant, so the paper is of great interest.
While the reviewer asks whether the substantive issue, that is, choosing between tax and subsidy is correct. Since "When the government levies carbon emission taxes, the production costs of companies will increase. Firms may pass on production costs to consumers and reduce social welfare", the authors have not taken into account that subsidies are money that the consumer It has given the state and it redistributes it in subsidies instead of in other aspects such as education.

Leaving aside this fundamental issue, which would invalidate the argumentation of the work, there are other minor aspects that should be corrected.

The objective of the work is diffuse in the introduction, so the reviewer recommends specifying it.

The results could be clarified, since some of the comments included in the conclusions could be included as the results. Although it is advisable to present them using a table or graph. Specifically, the implications of the values ​​of H, is somewhat messy.

The conclusions must be reworked, trying to show the strengths of the model.

Author Response

(The authors gave the same response as above.)

Reviewer 3 Report

The quality of the paper is very high - it covers an interesting and significant topic, uses sound methodology, and the results are grounded in the conducted analysis. There are some minor changes that could be made in order to increase the quality of presentation.

Authors use the abbreviation "ROA" - it can be misleading as "ROA" stands also for "return on assets" and is broadly used in this context. Some details about the methodology should be provided in the Introduction. The second paragraph of Section 1 is too long. Literature review could be divided into a few smaller parts, according to, e.g., the main topic. Line no. 195: the estimation of discount rate should be explained. It is unclear why Authors use the example of China in the Section 3 - some justification must be provided. Another comment: in my opinion assuming the constant relation between the GDP growth rate and carbon emission tax may be a too simplistic approach. Authors should provide some source or explanation of the values assumed in Table 1. Discussion of the results of analysis shown in Tables 2-5 may possibly be affected by the cause versus effect issues - this needs to be addressed. The final Section is poorly written. There are some unnecessary repetitions (e.g., the second sentence). Moreover, Authors formulate generalizations such as the threshold value based on merely one example (of China). Finally, the obtained results are not compared to the previous studies, the limitations of the analysis are not given, and there are no directions for future studies.

Author Response

(The authors gave the same response as above.)
